# Half a century of rising extinction risk of coral reef sharks and rays

C. Samantha Sherman [1,2,37] ✉, Colin A. Simpfendorfer[3,4,37], Nathan Pacoureau[5], Jay H. Matsushiba[1], Helen F. Yan [6,7], Rachel H. L. Walls [1], Cassandra L. Rigby[3], Wade J. VanderWright[1], Rima W. Jabado[3,8], Riley A. Pollom [9], John K. Carlson[10], Patricia Charvet [11,12], Ahmad Bin Ali[13], Fahmi [14], Jessica Cheok[1], Danielle H. Derrick[1], Katelyn B. Herman [15], Brittany Finucci[16], Tyler D. Eddy [17], Maria Lourdes D. Palomares [18], Christopher G. Avalos-Castillo[19,20], Bineesh Kinattumkara[21], María-del-Pilar Blanco-Parra [22,23,24], Dharmadi[38], Mario Espinoza[25,26], Daniel Fernando[27,28], Alifa B. Haque[29,30], Paola A. Mejía-Falla[31,32], Andrés F. Navia [32], Juan Carlos Pérez-Jiménez [33], Jean Utzurrum [34,35], Ranny R. Yuneni[36] & Nicholas K. Dulvy [1]

Sharks and rays are key functional components of coral reef ecosystems, yet many populations of a few species exhibit signs of depletion and local extinctions. The question is whether these declines forewarn of a global extinction crisis. We use IUCN Red List to quantify the status, trajectory, and threats to all coral reef sharks and rays worldwide. Here, we show that nearly two-thirds (59%) of the 134 coral-reef associated shark and ray species are threatened with extinction. Alongside marine mammals, sharks and rays are among the most threatened groups found on coral reefs. Overfishing is the main cause of elevated extinction risk, compounded by climate change and habitat degradation. Risk is greatest for species that are larger-bodied (less resilient and higher trophic level), widely distributed across several national jurisdictions (subject to a patchwork of management), and in nations with greater fishing pressure and weaker governance. Population declines have occurred over more than half a century, with greatest declines prior to 2005. Immediate action through local protections, combined with broad-scale fisheries management and Marine Protected Areas, is required to avoid extinctions and the loss of critical ecosystem function condemning reefs to a loss of shark and ray biodiversity and ecosystem services, limiting livelihoods and food security.

Coral reefs are amongst the most diverse ecosystems on the planet, harbouring more than one-third of the ocean's fish species[1]. Yet, they face some of the most intense and widespread threats of any ecosystem and are increasingly under threat globally due to climate change, poor water quality, and coastal development[2,3]. Overfishing, however, is the most immediate direct and indirect threat to most reefs and the cascading consequences for reef ecology and dependent coastal communities are only now being revealed. Coral reef fisheries directly support the livelihoods and food security of over half a billion people[4,5]. In turn, the footprint of coral reef fisheries is determined largely by proximity and size of human communities to reefs[6] and their reliance on these ecosystems[4]. However, this human footprint far exceeds the local reef productivity across many of the world's reefs[7]. These intense fisheries eliminate larger-bodied size classes and high trophic level

fishes, particularly sharks and rays[8]. Such functional losses distort ecological pyramids through mesopredator release[9], leading to ecosystem disruptions[8] and cascading changes down coral reef food chains that lead to declines in functionally important herbivores[10].

Chondrichthyans (hereafter, sharks and rays) are a phylogenetically diverse and ecologically important megafaunal lineage on coral reefs. There are 30 families, 59 genera, and 134 species of reef-associated sharks and rays[11,12], each with varying degrees of coral-reef association; from residents that spend their entire lives at one or a few reefs (e.g. Halmahera epaulette shark, *Hemiscyllium halmahera*[12]), partial residents that spend most of their time on reefs (e.g. Australian weasel shark, *Hemigaleus australiensis*[13]), and those transient passing through reef habitats (e.g. Javanese cownose ray, *Rhinoptera javanica*[11]). These species fill a range of ecological niches, including: filter feeders, benthic invertivores, resident piscivorous mesopredators, transient apex predators, and more[14,15]. As highly mobile predators, some species are important nutrient vectors and controllers of primary production[16], while others influence primary production through fear-induced trophic cascades[17]. Here are four examples of nutrient vectoring and cycling. First, grey reef shark (*Carcharhinus amblyrhynchos*) derives >80% of its diet from pelagic fishes then transfers these nutrients onto reefs[16]. Similarly, reef manta rays (*Mobula alfredi*) feed primarily on pelagic zooplankton and transfer these nutrients onto the reef during the day, acting as vectors for horizontal and vertical nutrient transport[18]. Stingrays are important ecosystem engineers, bioturbating large stretches of soft sediment while feeding or burying themselves in sand[19,20]. Finally, larger transient apex predators, like the tiger shark (*Galeocerdo cuvier*) mediate ecosystem structure and function of seagrass beds through fear-induced changes in grazing behaviour of turtles and dugongs[17,21], which may be a stronger effect than direct predation[22].

The warning signs of reef shark depletion first became apparent from space-for-time comparisons of recreational diver surveys of the Caribbean[23], comparisons across various levels of protection on the Great Barrier Reef[24], and in the remote atolls of the Indian and Pacific Oceans[25,26]. The most compelling evidence of widespread depletion comes from Baited Remote Underwater Video (BRUV) surveys of all shark species across 371 reefs, which found that a subset of sharks are functionally extinct at 20% of sites[27]. These conclusions relate to only a small fraction of reef shark and ray diversity; the survey recorded

59 species but 93% of the observations were of only the 10 most common sharks. The key question remains whether these local and regional surveys of the most common sharks forewarn of a global loss of coral reef shark and ray biodiversity and associated diminution of ecosystem services.

Here, we answer this question with the recently completed International Union for Conservation of Nature (IUCN) Red List of Threatened Species reassessment of sharks and rays[28]. Specifically, we provide a comprehensive assessment of the extinction risk of all 134 coral reef associated shark and ray species and compare their status with all other 4918 coral reef species assessed using IUCN Red List Categories and Criteria. Second, we identify the key threatening processes using (i) the IUCN threat classification scheme[28], combined with (ii) species-level[29] and (iii) national-level trait-based vulnerability analyses[30]. Third, we develop an IUCN Red List Index to track the progress toward international biodiversity targets over the past half century[31].

## Results
### Shark and ray extinction risk
We estimate that two-thirds of coral reef sharks and rays are threatened with extinction, based on the observed number of threatened species combined with the estimated number of Data Deficient (DD) species that are likely to be threatened (Critically Endangered (CR), Endangered (EN), or Vulnerable (VU)). Of the 134 species assessed, 79 (59.0%) are in a threatened category with 14 (10.5%) CR, 24 (17.9%) EN, and 41 (30.6%) VU. A further 18 (13.4%) species are Near Threatened (NT), 28 (20.9%) Least Concern (LC), and nine are DD (6.7%). Assuming that DD species are threatened in the same proportion as non-DD species[28], we estimate that 59.0%–65.7% of coral reef sharks and rays are threatened (mean 63.2%; n = 134; Fig. 1). This is the first assessment for 12.7% (n = 17 of 134) of sharks and rays, and these species arose from recent changes in taxonomic concept (Supplementary Data 1). Overall, most species (70.1%, n = 94), whether threatened or otherwise, exhibit a decreasing population trend; only 10.4% (n = 14) are stable and the population trend is unknown for 18.7% (n = 25). Only a single species - the bluespotted ribbontail ray (*Taeniura lymma*), is increasing globally, based on BRUV abundance estimates from across its range[32]. Overall, rays (range 64.3%–72.9%; n = 70) are more threatened than sharks (53.1%–57.8%; n = 64).

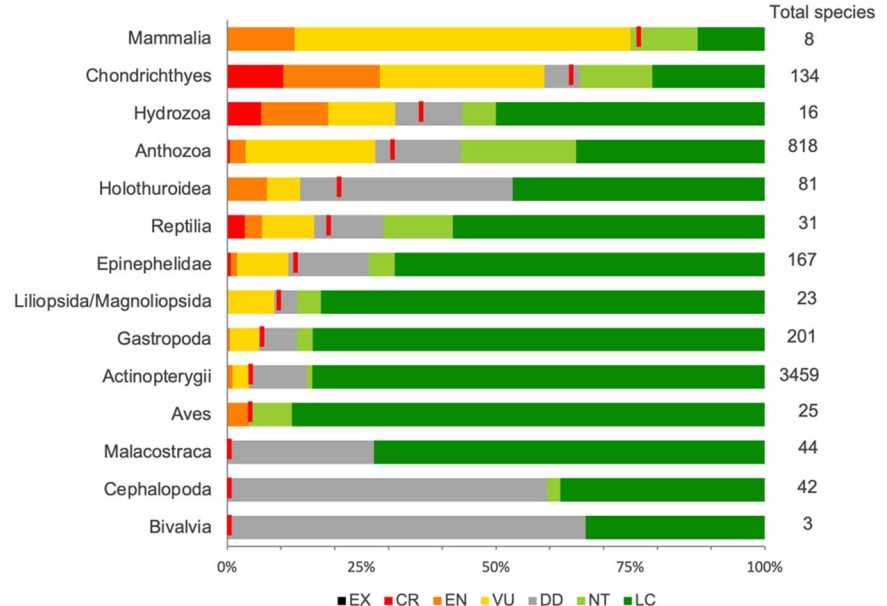

**Fig. 1 | The percent of coral reef species in the IUCN Red List categories.** Red lines indicate the best estimate of threat assuming all Data Deficient (DD, grey shade) species faced a similar level of threat to the data-sufficient species in the taxon. Extinct (EX), Critically Endangered (CR), Endangered (EN), Vulnerable (VU), Near Threatened (NT), and Least Concern (LC). Source data are provided as a Source Data file.

## Threat comparison to other taxa

Sharks and rays are the second most threatened group of the 4918 species found on coral reefs that have been assessed against Red List Categories and Criteria. Only marine mammals, comprising eight species of dolphins and sea-cows, are more threatened (Fig. 1). Eight taxa range from 4–31% of species in threatened categories, compared to 59% of sharks and rays, while the other three classes (Bivalvia, Cephalopoda, and Malacostraca) have no threatened species, based on those currently evaluated (Fig. 1). The high level of threat for coral reef sharks and rays is almost twice that of the whole Class Chondrichthyes, for which the estimated threat level is 38.5% for all 1199 species[28]. Coral reef sharks and rays are more at risk than all coastal sharks and rays, in which just over half (50.9%, $n = 296$ of 582)[28] are threatened, but less threatened than oceanic pelagic sharks and rays (77.4%, $n = 24$ of 30)[33].

## Overfishing is the main threat

Three lines of evidence support widespread overfishing as the principal cause of the high threat level of coral reef sharks and rays: (1) IUCN threat classification, (2) species trait-based analysis, and (3) nation trait-based analysis. For all species with available threat information (100%, $n = 132$) fishing was identified as a threat mainly due to unintentional catch in both small and large-scale fisheries (Fig. 2a). Fishing was the sole threat for 75 species (57%). Targeted fishing was a threat for over two-thirds of species (68%) and unintentional catch was a threat for almost all species (98%) (Supplementary Table 1). Other

threats that act in conjunction with fishing include: climate change (36% of species; $n = 47$ of 132), habitat loss and degradation [36% total; due to residential and commercial development (29%), agriculture and aquaculture (8%), and human disturbance (8%)], and pollution (9%). Uses of reef sharks and rays were available for 109 species. They are commonly used for human consumption (98%, $n = 107$ of 109) and body parts are fashioned into apparel/accessories (e.g. leather from ray skin; 23%); sharks are predominantly traded internationally for their fins and meat, and rays for their meat and skins, which are turned into leather[34,35]. Other common uses include aquarium display (27%), food for animals (19%), and medicines (11%).

## Species traits explain extinction risk

Migration behaviours and trophic level were important in explaining risk. Transient species had the highest overall threat level (76%), followed by partial residents (67%), while resident species had the lowest threat (44%; Fig. 2b). Resident species, including mainly benthic sharks and rays, often hide within the structures on a coral reef and are not easily caught by fishing gears; as opposed to partial residents or transient species, which transit through soft-bottom habitats which are comparatively easy locations to operate trawls and gillnets effectively[36,37]. Partial residents did have a higher proportion of Critically Endangered species than the other two groups, possibly due to high intrinsic sensitivity and small geographic range. These CR partial resident species tended to be larger bodied with long generation

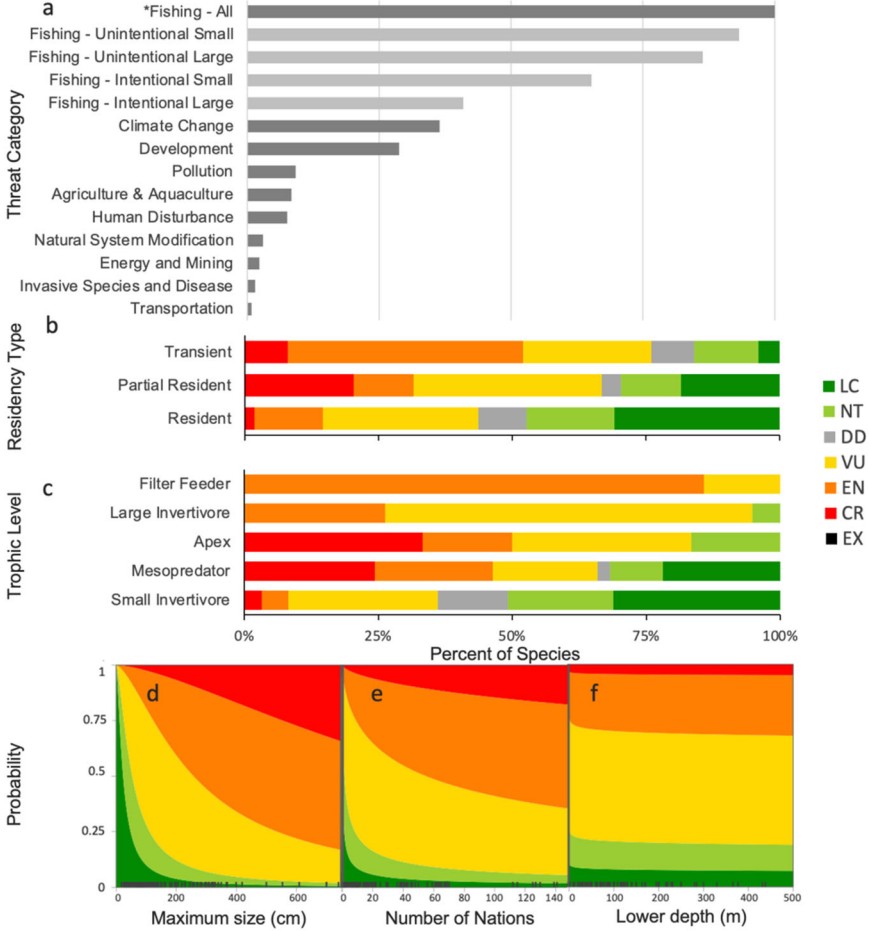

**Fig. 2 | High extinction risk of coral reef sharks and rays due to exposure to overfishing and other threatening processes combined with sensitive life histories. a** IUCN Threat classification for coral reef species. Percent of coral reef shark and ray species in each IUCN category by **b** residency patterns, and **c** trophic positions. The effects of **d** maximum body size, **e** geographic range size (indexed by number of nations a species occurs in), and (**f**) lower depth limit on the probability that a data-sufficient reef shark or ray species is listed as either Critically Endangered (CR), Endangered (EN), Vulnerable (VU), Near Threatened (NT), or Least Concern (LC) based on cumulative link mixed-effects models. Source data are provided as a Source Data file.

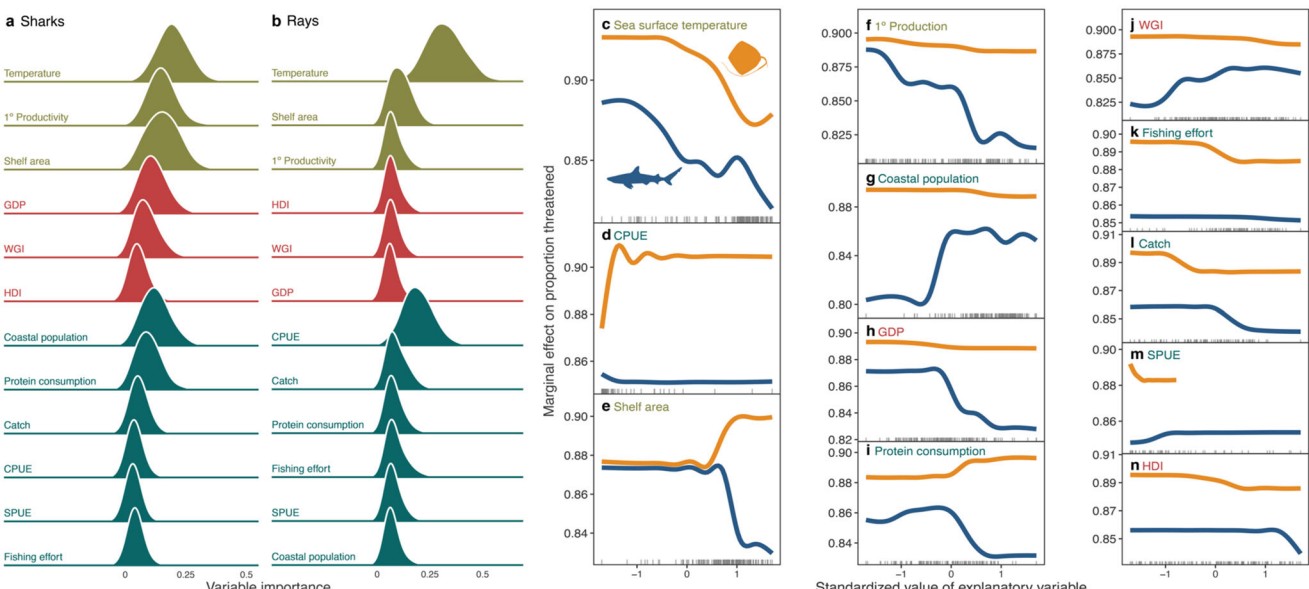

**Fig. 3 | National attributes also explain extinction risk.** Variable importance for explaining proportion of threatened species of all covariates across 1,000 boot-strapped boosted regression tree models for **a** sharks and **b** rays. The marginal effect **c**–**n** of each variable on the proportion of threatened species for rays and sharks ordered by variable importance. Explanatory variables were all scaled and centred around zero for presentation purposes; raw values were used in the analysis. Explanatory variables include: **c** sea surface temperature, **d** catch-per-unit-effort (CPUE), **e** shelf area, **f** primary production, **g** coastal population, **h** Gross Domestic Product (GDP), **i** marine protein consumption, **j** World Governance Index (WGI), **k** fishing effort, **l** catch, **m** sightings-per-unit-effort (SPUE), and **n** Human Development Index (HDI).

lengths or occurred in fewer than five nations. The filter feeders (low trophic position) were threatened, likely due to other attributes including their transient nature and large size. Overall, however, the higher trophic level species were significantly more threatened, based on an ordinal model (z-value(df=120)=2.75, $p = 0.006$) (Fig. 2c; Supplementary Data 2 and Supplementary Table 2).

Our results demonstrate that extinction risk is related to intrinsic life history sensitivity and indirect measures of the relative exposure to fishing mortality. The top-ranking ordinal logistic regression model revealed risk was greater in larger-bodied species (z-value(120)=4.35, $p < 0.0001$, coefficient estimate=2.16, 95% confidence interval (CI) = 1.17, 3.16), those occurring in a greater number of nations (z-value(120) =2.53, $p = 0.012$, coef.=1.24, 95% CI = 0.26, 2.23) and threat probability decreased slightly for species that occurred to depths greater than 100 m, suggesting depth refuge from threats (z-value(120)=−3.16, $p = 0.002$, coef.=−1.25, 95% CI = −2.04, −0.46; Fig. 2d–f and Supplementary Fig. 1). Relatively few non-threatened species had a maximum body size greater than 200 cm (24%, $n = 13$ of 55), while most (87%, $n = 20/23$) species over 300 cm maximum body size were Endangered or Critically Endangered. Geographic range varied from 1270 to 171,329,721 $km^2$ and the number of nations a species occurred in varied from one to 148. The more nations a species occurs in, the more fisheries management regimes it may encounter. Species found in 10 or fewer nations had a relatively low risk (43%, $n = 29/68$) compared to the high risk faced by widespread species found in 50 or more nations (95%, $n = 21/22$). Residency type and trophic level traits significantly contributed to the status of sharks and rays on coral reefs, however, correlation between traits and maximum linear dimension (measured either as total length or disc width, see Methods) or number of nations excluded it from the final model (Supplementary Table 3 and Supplementary Fig. 2).

Species with larger maximum body sizes often have lower intrinsic rates of population growth, less capacity to withstand fishing mortality, and are typically the most commercially valuable[38,39]. However, the connections to extinction risk and attribution to overfishing are complicated because larger-bodied species also tend to have larger

geographic ranges[40]. Yet, larger geographic ranges might not reduce extinction risk because mobile wide-ranging species transit through more jurisdictions and spend less time in protected areas, thus increasing their exposure to a wider range of fisheries and more frag-mented management[41]. Threat level is determined by the interaction of intrinsic sensitivity, such as body size, with a threatening process, which we indexed with depth and geographic range as a measure of the accessibility of a species to fisheries and fragmented management[29].

## National attributes also explain extinction risk

We found that the national attributes associated with variations in extinction risk differed between coral reef sharks and rays. For both sharks and rays, the national percentage of globally threatened species was associated mainly with abiotic factors related to population size [sharks: 50.2%, rays 46.6% of the summed average variable importance (AVI)], followed by fishing pressure (27.1% and 43.5%, respectively), and management capacity (22.6% and 9.9%, respectively; Fig. 3a, b; Supplementary Notes; Supplementary Fig. 3).

Sea surface temperature had the highest overall association with shark and ray extinction risk, with a greater importance for rays (32.4% AVI) than sharks (19.8% AVI; Fig. 3c), possibly indicating a latitudinal effect, where there are fewer threatened species in more tropical waters. Primary production was the second most important variable for sharks with a decreased extinction risk with increasing primary pro-duction (15.3% AVI), possibly due to increased population sizes resulting from carrying capacity and prey availability (Fig. 3f)[26]. National catch-per-unit-effort (CPUE) of reef-associated species was the second most important variable for rays (19.2% AVI) such that at very low CPUE, rays had a lower extinction risk and a slight increase in CPUE was associated with a rapid increase in extinction risk (Fig. 3d). The extinction risk of sharks was greatest in nations with larger coastal human populations, and this was the most important predictor for sharks from the fishing pressure variables (12.3% AVI), although it had little importance for rays (2.8% AVI; Fig. 3g). The level of marine protein consumption was moderately important for sharks and rays (9.1 and 6.4%, respectively), but had a counterintuitive relationship with their extinction risk; shark

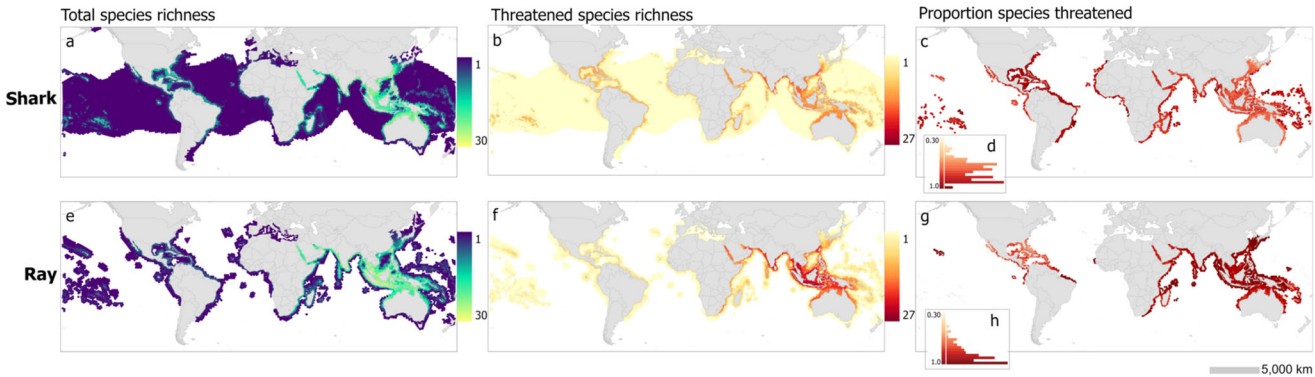

**Fig. 4 | Global shark and ray species richness, threat, and proportion threatened by group. a, e** species richness by group. **b, f** number of threatened species by group. **c, g** threatened species as a proportion of total richness (for cells with >5 species) by species group. **a–d** sharks and **e–h** rays. Scale bar: 5000 km. **d, h** histograms represent the number of grid cells containing different percentages of threatened species.

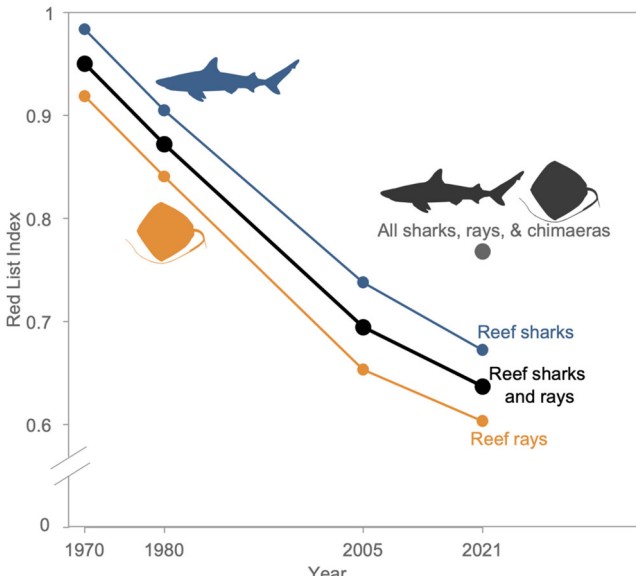

**Fig. 5 | Increase in extinction risk of coral reef sharks and rays over the past half century.** Global Red List index (RLI) for the 125 data-sufficient reef sharks and rays combined, and each separately, estimated in 1970, 1980, 2005, and 2021, and compared to the RLI for all shark and ray species. An RLI value of 1 indicates that all species in the taxa are Least Concern and an RLI value of 0 indicates that all species are Extinct. Source data are provided as a Source Data file.

extinction risk decreased with an increase in marine protein consumption, whereas ray risk increased (Fig. 3i). World Governance Index (WGI) and Human Development Index (HDI) did not have strong influence on reef shark or ray extinction risk (Fig. 3j, n), indicating that even nations with higher governance capacity are not always translating this into conservation success for sharks and rays.

### Where are reef species threatened?

Separately, both shark and ray richness on coral reefs are highest in northern Australia and Southeast Asia, with up to 26 and 30 species per 23,322 km² grid cell, respectively (Fig. 4a, e; Supplementary Fig. 4). The highest proportion of threatened species, however, differs such that sharks have a higher threat proportion in the western Atlantic (Fig. 4c) and rays had the highest proportion threatened throughout Asia and southeast Africa, with over 75% of ray species threatened in these areas (Fig. 4g). Species' status are applied homogeneously throughout their entire range, despite known differences in status among nations of occurrence. Therefore, the percent of species threatened appears high

in the Pacific Islands and other remote locations despite studies showing that these locations often have healthy populations of sharks[26,27]. Indeed, a beta regression model proved the proportion of reef sharks and rays threatened did not have a significant relationship to local relative abundance based on the results in MacNeil et al. (2020), which included shark and ray surveys on coral reefs in 52 nations (z-value(51)=−0.914, $p = 0.361$; Supplementary Fig. 5). Locations like the Pacific Islands, with higher abundances of reef sharks and rays act as refuges for threatened, widespread species and are important for regional and global conservation efforts (purple quadrant, Supplementary Fig. 5). Other nations with low abundances and high threat should be the focus for conservation capacity building[27,42] (yellow and red quadrants, Supplementary Fig. 5). Thus, the maps should not be used in isolation to identify locations for conservation action, but rather used with information on local abundance (e.g., MacNeil et al. 2020).

### Red list index

The Red List Index (RLI) is a measure of extinction status across a group of species, where a value of 1 indicates all species are Least Concern, and a value of 0 indicating all species are extinct. Risk of extinction has substantially increased since 1970 for the 125 coral reef shark and ray species included in this study that were not assessed as DD. The RLI of all coral reef sharks and rays declined from a retrospective estimate of 0.95 in 1970 to 0.64 in 2021. Coral reef sharks and rays currently have a lower RLI than the whole Class Chondrichthyes (RLI = 0.77, $n = 1199$) (Fig. 5). Reef rays ($RLI_{2021} = 0.60$) are more at risk of extinction than reef sharks ($RLI_{2021} = 0.67$), but their RLI trends are similar, with sharks and rays declining faster between 1980 and 2005 and slightly less so between 2005 and 2021 (Fig. 5). Most of the RLI decline for reef sharks and rays occurred prior to the 2005 assessment, likely due to a boom in coral reef fisheries in the 1960s and 70 s and collapse by the early 2000s[7,43]. As a result, regions like Southeast Asia, where the highest species richness occurs, were already experiencing overfishing by the 1980s[43]. The 2021 reef shark and ray RLI value is lower than any other coral reef taxa, including mammals, which have a higher percent of threatened species (Fig. 1).

## Discussion

We show that the global extinction risk of sharks and rays, as a percentage of threatened species, on coral reefs is almost double that of all 1199 sharks and rays[28]. Overfishing is the major threat reported and the main cause for population declines, causing dramatic declines in a very short space of time (i.e., the past 50 years). This work builds on previous work by MacNeil et al. (2020)[27] showing that a smaller number of common shark species were functionally extinct on

approximately 20% of surveyed coral reefs. Our analysis extends this study by including the complete list of coral reef sharks and rays, and by demonstrating that rays, as well as sharks, have high levels of extinction risk. With many threatened species, including 14 that are Critically Endangered, it is likely that the number of reefs where sharks and rays are functionally extinct will increase unless urgent action is taken to address threats. With sharks and rays playing important functional roles in coral reef ecosystems[14], and their presence or absence affecting both the abundance and behaviour of other species[9], there will be growing ecological consequences for coral reefs, many of which will be hard or impossible to reverse[7,15].

To achieve recovery, fishing mortality from subsistence, artisanal, and industrial fisheries needs to be controlled, with the best way forward being science-based fisheries management (both traditional and 'western') with strong enforcement[44,45] and increased use of, well-implemented and enforced, marine protected areas (MPAs)[46]. Additionally, due to the large spatial distribution of many coral reef species, regional level management should be implemented to address species' status differences between neighbouring countries. For example, in Central and South America, four nations have created the Eastern Tropical Pacific Marine Corridor (CMAR) to preserve biodiversity through protecting marine corridors, which will benefit migratory and transient species[47]. Despite climate change being a major global threat to coral reefs[2], and a threat to several coral reef shark and ray species[48], it is not yet the most immediate threat that needs to be addressed to reverse the declines in shark and ray populations. In particular, the use of MPAs to successfully recover coral reef shark populations has been demonstrated both in theory[49] and in practice[24,50]. The specific implementation of management, either through fisheries regulations or MPAs, depends on the context in which it occurs (e.g., Mizrahi et al. 2019[51]). In many developing nations, where coral reef fisheries play important food security and livelihood roles, management responses will be different to those in developed nations where alternative livelihoods and protein sources are available[45]. In addition to fisheries management and MPA implementation, the following recommendations are required to ensure long-term survival of coral reef sharks and rays: (1) capacity development to enable implementation of conservation actions, including fisheries monitoring and enforcement[42] and through development of Sustainable Development Goals and Convention on Biological Diversity targets (IPBES 2019)[52,53]; (2) improved education and diversification of rural livelihoods in nations with overexploited reefs to reduce fishing pressure on threatened species[54,55]; and (3) better use of trade controls to address international demand (e.g., Convention on International Trade in Endangered Species of Wild Fauna and Flora, CITES), particularly with recent increases in the shark meat and ray skin trade, which includes a large suite of reef species[56]. These recommendations are currently being applied to varying degrees and with varying success. The first two also address climate change threats to coral reefs by increasing resilience by maintaining abundance and conserving biodiversity[57,58], and reducing human reliance on reefs through alternative livelihoods[59]. Without broad-scale action to improve the status of sharks and rays on coral reefs, the declines in global populations reported here will continue, with increasingly dire consequences for the ecosystem health of coral reefs and coastal communities that rely on them.

## Methods

### IUCN red list assessments

The IUCN Red List of Threatened Species, hereafter referred to as the IUCN Red List, provides the most widely-used approach for assessing extinction risk[60]. Species evaluated are classified according to the IUCN Categories and Criteria (Version 3.1) following the Guidelines for Using the IUCN Red List Categories and Criteria and these categories range from Least Concern to Extinct, with those that have an elevated risk of extinction considered threatened (Vulnerable, Endangered, or Critically Endangered)[61]. Extant species assessed as not having an elevated risk of extinction are placed in the categories of Least Concern or Near Threatened (i.e., those that may soon be eligible for a threatened category). The coral reef species were assessed as part of a nine-year effort to reassess all 1199 species, through 17 taxonomic and thematic workshops, with 353 participants from 244 unique experts and/or members of the IUCN Species Survival Commission (SSC) Shark Specialist Group (SSG) from 71 countries and territories (Supplementary Data 3). The collation of information, assessment, peer-review, and consultation process are detailed in Dulvy et al. (2021). Some of the coral reef shark assessments required estimation of population reduction using hierarchical Bayesian state-space methods and space-for-time substitution methods and these are detailed in the supplementary files associated with the Red List Assessments[28,62].

### Species list and attributes

We compiled a list of 134 coral reef associated chondrichthyans (70 rays, 64 sharks). We included all species listed on the IUCN Red List that included 'coral reef' as a habitat ($n = 126$) as of April 17, 2021. We removed 12 of those species for various reasons (Supplementary Table 4), leaving 114. We then added an additional 20 species that also occur at coral reefs at some point during their lives, based on expert knowledge (Supplementary Table 5). Species were split based on their residency pattern on coral reefs, ranging from 'resident,' 'partial resident,' and 'transient'. Residents are those species that spend their entire lives on coral reefs (e.g., bluespotted lagoon ray, *Taeniura lymma*). Partial residents include species that use coral reefs for most of their lives, but use other habitats as well (e.g., bowmouth guitarfish, *Rhina ancylostoma*). Finally, transient species are those that use reefs occasionally but spend most of their lives in other habitats (e.g., spotted eagle ray, *Aetobatus narinari*) (Supplementary Data 2). We additionally categorized the species based on their trophic level into filter feeders, small invertivores, large invertivores, mesopredators, and apex predators (Supplementary Data 2). Large invertivores were classified as rays with a maximum size greater than 100 cm disc width. For statistical analyses, these are coded in an ordinal manner from 1 to 3 for residency type (1 for residents to 3 for transients) and from 1 to 5 for trophic level (1 for filter feeders up to 5 for apex predators). Maximum size values were obtained from the published species' Red List Assessments.

Threats to sharks and rays were analysed at three levels: 5 Biological Resource Use, 5.4 Fishing and harvesting aquatic resources), and 5.4.1 Intentional mortality (human use - subsistence/small scale))[63]. They were determined based on coded threats in the IUCN Red List assessment for each species with the exception of climate change[64]. Climate change was also included as a threat if it was mentioned in the text of the 'threats' section of an assessment regardless of being included in the coded threats. For example, the Arabian carpetshark (*Chiloscyllium arabicum*) did not have 'climate change' included as a coded threat, however, it was discussed in the threat section text and therefore included[65].

### Generation length

Generation lengths were taken from IUCN Red List assessments when available, where it is calculated as the mid-point between the age-at-maturity and longevity[28]. Generation lengths were either taken from published age-at-maturity and longevity for the species, directly from the species' IUCN Red List assessment, or estimated using proxy data from related species that have known age-at-maturity and longevity (Supplementary Data 2).

### Extinction risk and other reef taxa

To compare the status of sharks and rays to other coral reef taxa, the IUCN Red List website (www.iucnredlist.org) was searched on February 4, 2022 for all animal Classes that included at least a single species with 'coral reef' listed as a habitat as per their respective IUCN Red List assessment. The numbers of species within each Class and each

IUCN Red List status were recorded. Additionally, we compare reef elasmobranchs to the family Epinephelidae (Coral Groupers), which were recently reassessed and consist mostly of coral reef species[66]. We included a total of 4918 species (4747 through the IUCN website, additional 167 by including all Epinephelidae[66], and 134 sharks and rays [this study]). To best estimate the true percent threatened for each species group considering DD species, we considered that DD species were equally as threatened as data-sufficient species as per IUCN guidelines (Eq. 1; www.iucnredlist.org/resources/summary-statistics).

$$\text{Threatened(estimated \%)} = \frac{(CR + EN + VU)}{(\text{total assessed} - DD)} \qquad (1)$$

### Species traits and threat level

Each species' geographic range was calculated as the total two-dimensional area of the species' distribution as mapped in each species' respective IUCN Red List assessment. To determine the number of management schemes a species fell under, the number of nations was included as a variable, calculated by determining the number of different Exclusive Economic Zones (EEZs) a species' distribution overlapped with.

Statistical models were conducted with R v.4.1.0 using relevant software packages[67]. The variables that could best predict extinction risk of a species were estimated using ordinal logistic regression models (ordinal package[68]). All numeric variables (species range, generation length, maximum linear dimension, number of nations, lower depth) and the ordinal values of residency type and trophic level were tested for correlation (Pearson's $r$; PerformanceAnalytics package[69]). Variables with correlation >0.7 were not included in the same models (Supplementary Fig. 2)[70]. As shark sizes are measured as total length and rays are generally measured as the disc width, models were tested both with and without maximum sizes nested in measurement type (i.e., total length and disc width). Models performed better when measurement type was not considered, therefore, models were run with the variable 'maximum linear dimension', which included both measurement types. For residency type, sharks and rays were split into 55 resident species, 54 partial residents, and 25 transient species (Supplementary Data 2). Due to the high degree of pairwise correlation between each of residency and trophic level traits with maximum body size, these could not occur in the same models with maximum body size and measures of geographic range, which were preferentially included in the top model based on AIC (Supplementary Table 3 and Supplementary Fig. 2). In addition to a null model, we ran 35 proposed models with different combinations of: generation length, residency type (both ordinal and categorical), trophic level (both ordinal and categorical), lower depth limit, maximum linear dimension, number of nations, and species range (Supplementary Table 3). All models were tested for variable collinearity using variance inflation factors (VIF), and only models with all variables returning values of <2 were further considered[71]. The number of nations a species occurs in was more important at predicting risk than the geographic range and was included in the final model. The most parsimonious model with an AIC value within 2 units of the lowest AIC was selected as the best performing model[72]. Coefficients were estimated and standardised for effect magnitude of variables in the top ordinal model (coefplot package[73]).

### National trait boosted regression tree analysis

Management typically occurs at the national level, therefore, we used a machine learning algorithm to model the percent of threatened species per nation against 11 national-level covariates as per Yan et al[30]. (Supplementary Data 4). We used gross domestic product (GDP (mean from 1960–2019) in USD; https://data.worldbank.org/indicator/NY.GDP.MKTP.CD), world governance index (WGI (mean from 1996–2018); https://databank.worldbank.org/source/worldwide-governance-indicators), and human development index (HDI (2019); http://hdr.undp.org/en/content/human-development-index-hdi) as indicators of governance effectiveness. To model both direct and indirect fishing pressure, we used coastal human population size, measured as the number of people living within 100 km of the coast within the distribution of coral reefs in 2020 (https://sedac.ciesin.columbia.edu/data/set/nagdc-population-landscape-climate-estimates-v3), and marine protein consumption (mean from 1961–2013 in g capita$^{-1}$ day$^{-1}$;[74]) to represent the reliance on marine fish products for dietary protein and economic stability. We also included fishing effort, total catch (units), sightings-per-unit-effort (SPUE), and catch-per-unit-effort (CPUE) of all coral reef associated species (all as mean values from 1950–2010)[7]. We included the 2020 annual mean sea surface temperature[75] and the 2020 annual mean primary productivity (measured as chlorophyll-a concentration[76]) as ecological indicators and included continental shelf area clipped to a bathymetry of 50 m within each nation's EEZ to the extent that coral reefs are found to denote total habitat availability[77].

We used a boosted regression tree (BRT) framework from the XGBoost package[78] to model the effect of governance, fishing, and ecological indicators on the percent of threatened species per nation. We separated sharks and rays for analyses due to their differing ecologies and responses to threats. BRTs are a form of machine learning whereby a boosting algorithm is used to fit many decision trees while minimising a loss function. Consequently, BRTs can handle non-linear data and complex interactions and are not limited by collinearity nor missing data[79]. The correlation (Pearson's $r$) among explanatory variables were all below 0.8, and we log-transformed coastal population, GDP, protein consumption, shelf area, primary productivity, fishing effort, catch, SPUE, and CPUE to improve their distributions[79]. We first used a tuning step to select hyperparameters by varying the learning rate (eta), maximum loss reduction (gamma), the maximum tree depth (max_depth), and the subsample ratio of the training instance (subsample) to minimise the root mean squared error. To suit the 0-1-inflated-beta distribution of our data, our final model was fit with a logistic loss function with the following hyperparameters (sharks, rays): eta = 0.5, 0.2; gamma = 0.5, 0.7; max_depth = 15, 15; subsample = 0.9, 0.3. With this tuning step, we were able to reduce the root mean squared error from 0.093 to 0.078 for sharks and from 0.069 to 0.063 for rays.

We used a form of cross-validation to fit our final model, where we split our data into an 80–20% training-test set split. Due to the stochastic model building process of BRTs, we randomised our data before each iteration and bootstrapped the models for 1000 iterations. We measured the biases of each model by subtracting the predicted percent threatened from the measured value in the test set, whereby a value close to zero would denote minimal prediction bias. We also extracted the relative importance and calculated the marginal effect of each variable for each iteration. All analyses were conducted using the XGBoost package v.1.4.1.1[78] in R v.4.1.0[67].

The average bias of the models across 1000 bootstrapped iterations was $-2.5 \times 10^{-3}$ (95% confidence interval: $-4.0 \times 10^{-3}$ to $-1.1 \times 10^{-3}$) for sharks and $-1.0 \times 10^{-3}$ ($-2.0 \times 10^{-3}$ to $1.0 \times 10^{-4}$) for rays, indicating the models has a negligible bias. The root mean squared error across all iterations was 0.089 (0.088 to 0.089) for sharks and 0.089 (0.089 to 0.089) for rays (Supplementary Fig. 3).

### Mapping

To determine global patterns of richness and threat, species maps were produced for all species combined, and sharks and rays separately. All maps were prepared using ArcGIS Pro 2.7.0[80]. We spatially joined the polygons representing species ranges to a hexagonal grid of individual units (cells) that retain their shape and area (~23,322 km²) throughout the globe to generate a species count in each cell. We then projected maps to the Patterson projection. Maps

were truncated to 60° latitude because no data existed outside these areas with the exception of Iceland and Alaska, where there has been at least one reported observation of a tiger shark (*Galeocerdo cuvier*). The tiger shark is reported as a seasonal vagrant and not a resident of these nations, therefore, their removal from the map did not affect the tiger shark's resident distribution[81]. For the proportion of threatened species maps, only cells with greater than five species were included to avoid disproportionately high threat in cells with very few species.

### Comparison of threat status and global FinPrint BRUVS abundance estimates

BRUVS estimates of MaxN (highest count of species in view at once) were downloaded from the publicly available data in the MacNeil paper[27] at (https://doi.org/10.1038/s41586-020-2519-y). Sightings-per-unit-effort was calculated by summing the total MaxN per nation and dividing it by the total hours of footage analysed. Species lists per nation were created using the IUCN Red List assessment geographic ranges. The proportion of species threatened was calculated by dividing the total number of CR, EN, and VU species in a nation over the total number of coral reef sharks and rays in the nation. A beta regression linear model was run to compare the proportion of coral reef sharks and rays threatened in each nation and SPUE of coral reef sharks from the MacNeil paper (betareg package[82]).

### Red list index

The Red List Index (RLI) values from subsequent assessments show changes in relative extinction risk of various taxa over time[31,53]. The RLI was calculated by experts retrospectively assigning an IUCN Red List category to each species for the years 1970, 1980, and 2005 based on current status and known history of threat severity. In particular, we used reconstructed catch data from the Sea Around Us, which includes the dates, locations, and species groups exploited. With this information, along with an understanding of species-specific traits and response to fishing, we could estimate that if fishing pressure increased throughout the 1970s and reached a peak in the early 2000s, the species in question was likely Least Concern in 1970, possibly Near Threatened by 1980, and likely in a threatened category by 2005. We built a picture of the history of fishing pressure using the Sea Around Us data and worked backwards from the current category to determine if the previous category (e.g., from 2020 to 2005, then 2005 to 1980, then 1980 to 1970) was likely to be: (i) the same, (ii) better by one, or (iii) two categories. Thus, the range of possible choices is highly constrained at each timestep. Though some assessments were completed around the year 2005, using our current understanding of threats and trends, these may not have accurately represented the IUCN Red List status at the time. Therefore, the 2005 RLI may differ from published IUCN Red List assessments. Eight species underwent taxonomic revision and thus, the initial IUCN Red List assessment may no longer be relevant. However, for these eight species, the old taxonomic concept covers most of their range and, therefore, these were not considered to have previously been Not Evaluated (Supplementary Table 6). Once IUCN Red List categories were assigned, the RLI for each year ($t$) was calculated by multiplying the number of species included ($s$) in each IUCN Red List category by their respective category weights ($W_c$; 0 for LC, 1 for NT, 2 for VU, 3 for EN, 4 for CR, and 5 for EX). As per Eq. 2, the product was then summed and divided by the maximum possible product (number of species ($N$) multiplied by the maximum weight (i.e. $W$=5)), and subtracted from 1 to achieve a final index ranging between 0 (meaning all species are Extinct) and 1 (meaning all species are Least Concern[83]):

$$\text{RLI}_t = 1 - \frac{\sum_s W_{c(t,s)}}{W_{\text{EX}} * N} \qquad (2)$$

### Reporting summary

Further information on research design is available in the Nature Portfolio Reporting Summary linked to this article.

## Data availability

IUCN Red List of Threatened Species assessments are publicly available with links to specific assessments in Supplementary Data 1. All species traits used in analyses are available in Supplementary Data 2. All covariate values used for the national trait boosted regression tree analysis and the source of these values are available in Supplementary Data 4. Source data are provided with this paper.

## Code availability

Code used to reproduce the analysis can be accessed at https://github.com/sammsherman27/CoralReefSharkRayIUCN and at https://doi.org/10.5281/zenodo.7267904[84].

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

## Acknowledgements

We thank all members of the IUCN Species Survival Commission Shark Specialist Group and other experts who contributed to the data collation and, in particular, R. Barreto, A. Cevallos, C. Dudgeon, D. Ebert, A. González, P.M. Kyne, A. Maung, J.M. Morales-Saldaña, L. Seyha, A. Sianipar, D. Tanay, and V.Q. Vo (Supplementary Data 3). We thank W.W.L. Cheung and V.W.Y. Lam for their contributions to the coral reef catch and effort data. We thank C.M. Pollock and C. Hilton-Taylor for quality-controlling data submission through the IUCN Red List. The scientific results and conclusions, as well as any views or opinions expressed herein, are those of the author(s) and do not necessarily reflect those of institutions or data providers. This project was funded by the Shark Conservation Fund, a philanthropic collaborative pooling expertise and resources to meet the threats facing the world's sharks and rays. The Shark Conservation Fund is a project of Rockefeller Philanthropy Advisors. This work was funded by the Shark Conservation Fund as part of the Global Shark Trends Project to N.K.D. and C.A.S. N.K.D. was supported by Natural Science and Engineering Research Council Discovery and Accelerator Awards and the Canada Research Chairs Program.

## Author contributions

C.S.S., C.A.S., and N.K.D. conceptualized the paper and the analysis. C.S.S., C.A.S., N.P., J.M., H.F.Y., R.W., and N.K.D. conducted the statistical analyses and visualised the data. J.M., J.C., D.D., and K.H. created and compiled species maps. C.A.S. and N.K.D. acquired the funding. C.S.S., C.A.S., and N.K.D. wrote the first draft. C.S.S., C.L.R., W.J.V., R.P., C.A.S., J.C., P.C., R.J., A.B.A., F., C.G.A.C., B.K., M.P.B.P., D., M.E., D.F., A.B.H, P.A.M.F., A.F.N., J.C.P.J., J.U., R.Y., and N.K.D. contributed significantly to Red List assessments that were used in this paper. T.E. and M.L.D.P. provided catch and effort data for coral reefs. All authors contributed to writing the manuscript.

## Competing interests

The authors declare no competing interests.

## Additional information

[1]Earth to Ocean Research Group, Biological Sciences, Simon Fraser University, 8888 University Drive, Burnaby, BC V5A 1S6, Canada. [2]TRAFFIC International, Cambridge, UK. [3]College of Science and Engineering, James Cook University, 1 James Cook Drive, Townsville, QLD 4811, Australia. [4]Institute of Marine and Antarctic Studies, University of Tasmania, Hobart, TAS 7000, Australia. [5]Department of Fish and Wildlife Conservation, Virginia Polytechnic Institute and State University, 310 West Campus Drive, 100 Cheatham Hall, Blacksburg, VA 24061, USA. [6]Research Hub for Coral Reef Ecosystem Functions, College of Science and Engineering, James Cook University, 1 James Cook Dr, Townsville, QLD 4811, Australia. [7]ARC Centre of Excellence for Coral Reef Studies, James Cook University, 1 James, Townsville, QLD 4811, Australia. [8]Elasmo Project, P.O. Box 29588 Dubai, Dubai, United Arab Emirates. [9]Species Recovery Program, Seattle Aquarium, 1483 Alaskan Way, Pier 59, Seattle 98101 WA, USA. [10]NOAA Fisheries Service, 3500 Delwood Beach Rd, Panama City 32408 FL, USA. [11]Programa de Pós-graduação em Sistemática, Uso e Conservação da Biodiversidade (PPGSis - UFC), Acesso Público, 913 - Pici, Fortaleza, Paraná 60020-181, Brazil. [12]Programa de Pós-graduação em Engenharia Ambiental (PPGEA - UFPR), Avenida Coronel Francisco Heráclito dos Santos, 210 - Centro Politécnico, Curitiba, Paraná 81531-970, Brazil. [13]Marine Fishery Resources Development and Management Department, Southeast Asian Fisheries Development Center, Jalan Pelabuhan LKIM, Chendering, Kuala Terengganu, Terengganu 21080, Malaysia. [14]Research Center for Oceanography, National Research and Innovation Agency - Indonesia, Jalan Pasir Putih Raya No.1 Ancol Timur, Jakarta Utara, DKI Jakarta 14430, Indonesia. [15]Georgia Aquarium, 225 Baker St. NW, Atlanta 30313 GA, USA. [16]National Institute of Water and Atmospheric Research (NIWA), Hataitai, Wellington 6011, New Zealand. [17]Centre for Fisheries Ecosystems Research, Fisheries & Marine Institute, Memorial University of Newfoundland, 155 Ridge Road, St. John's, NL A1C 5R3, Canada. [18]Sea Around Us, Institute for the Oceans and Fisheries, University of British Columbia, 2202 Main Mall, Vancouver, BC V6T1Z4, Canada. [19]Fundación Mundo Azul, km 21-22 Finca Moran, Villa Canales 01065, Guatemala. [20]Centro de Estudios del Mar y Acuicultura, Universidad de San Carlos de Guatemala, Ciudad universitaria zona 12, Guatemala City 01012, Guatemala. [21]Zoological Survey of India, Marine Biology Regional Centre, 30, Near Hotel Sangeetha Santhome High Road, Chennai, Tamil Nadu 600028, India. [22]Consejo Nacional de Ciencia y Tecnología, Av. Insurgentes Sur 1582, Crédito Constructor, Ciudad de México, Ciudad de México 03940, México. [23]Universidad Autónoma del Estado de Quintana Roo, División de Desarrollo Sustentable, Blvd. Bahía s/n, Del Bosque, Chetumal, Quintana Roo 77019, México. [24]Fundación Internacional para la Naturaleza y la Sustentabilidad A.C., Calle Larún, Mzn 75 Lt4, Andara, Chetumal, Quintana Roo 77014, México. [25]Centro de Investigación en Ciencias del Mar y Limnología, Universidad de Costa Rica, San Pedro de Montes de Oca, San José 2060-11501, Costa Rica. [26]Escuela de Biología, Universidad de Costa Rica, San Pedro de Montes de Oca, San José 2060-11501, Costa Rica. [27]Blue Resources Trust, 86 Barnes Place, Colombo, WP 00700, Sri Lanka. [28]Linnaeus University, Department of Biology and Environmental Science, Kalmar, SE 39182, Sweden. [29]Nature-based Solutions Initiative, Department of Zoology, University of Oxford, Research and Administration Building, 11a Mansfield Rd, Oxford, Oxfordshire OX1 3SZ, UK. [30]Department of Zoology, University of Dhaka, Dhaka 1000, Bangladesh. [31]Wildlife Conservation Society - WCS Colombia, Av. 5N # 22N-11, Cali, Valle del Cauca 760001, Colombia. [32]Fundación colombiana para la investigación y conservación de tiburones y rayas - SQUALUS, Calle 10A # 72-35, Cali, Valle del Cauca 760001, Colombia. [33]El Colegio de la Frontera Sur, Av. Rancho Polígono 2-A, Lerma, Campeche 24500, México. [34]Marine Wildlife Watch of the Philippines, G/F Bonifacio Ridge Building, 1st Avenue, Bonifacio Global City, Taguig 1634, Philippines. [35]Silliman University - Institute of Environmental and Marine Sciences, Dumaguete City, Negros Oriental 6200, Philippines. [36]WWF-Indonesia, Pemuda 1 No. 8, Denpasar, Bali 80224, Indonesia. [37]These authors contributed equally: C. Samantha Sherman, Colin A. Simpfendorfer. [38]Deceased: Dharmadi. ✉e-mail: sammsherman27@gmail.com

