## [Peer Review File · Nature Communications]

Half a century of rising extinction risk of coral reef sharks and raysREVIEWER COMMENTS

Reviewer #1 (Remarks to the Author):

Dear authors,

I have read with high interest the manuscript “Half a century of rising extinction risk of coral reef sharks and rays”. The article provides a deep insight into the conservation status of coral reefs’ sharks and rays, and nice explanations about the drivers of the high extinction risk of this group. I will provide some comments on some parts of the text that could be enriched, and others that may need to be clarified, but in general I find the manuscript well written, interesting and with quality enough to recommend its publication in the journal with a minor revision of some aspects.

I first realized that the authors based this study, or at least part of it, in the previously published one, “Overfishing drives over one-third of all sharks and rays toward a global extinction crisis”. I would like to thank the authors for overcoming my initial skepticism, as I thought the paper would just be a repetition of the previous one with the focus on coral reef species. On the contrary, the manuscript deepens in the conservation status of these species and provides a broad and complete overview on the causes of the observed threat patterns. I would like to highlight how different sources of information, very different among them, have been used to create a text easily readable as a whole, well-structured and with a clear narrative that allows the reader to understand most of the results observed, and they were obtained.

I would like to point out some aspects that I consider would help to clarify the comprehension of the manuscript.

Line 88: I cannot find any rule against references in the abstract in the Author Guidelines, but they do not usually appear there, I leave it to the editor’s decision.

Lines 116-120: I struggle to see the connection between these two sentences. Could you explain better how eliminating large predators leads to large herbivores disappearing too?

Line 177 (and also lines 405-425): Where does this number come from? I assume you are using IUCN Red List habitat 9.8, but this search lead to 4,880 species. How did you increase this number up to 4,918 species?

Line 179: Eight species of delphinids, and also sirenids?

Lines 211-213: Filter feeders, that are the most threatened trophic group in figure 2c, are not “species of higher trophic position”. It would be worthwhile to acknowledge this fact, considering the few filter feeders species, and their special condition as big and charismatic fishes (manta rays and whale shark), it is something that can be included.

Lines 227-228: This sentence is repeated from lines 219-220, it adds nothing new, please remove.

Line 530: Maybe could be useful to include a table with species retrospective assessments as a Supplementary Material.

Extended data Figure 5: How were the quadrants delimited? Arbitrarily, mean or median values, other reason? Please explain it.

Discussion: I enjoy the discussion of protective measures to prevent overfishing in coral reefs. How could these measures potentially be combined with the ones that are currently been applied? Since climate change has been traditionally classified as the main threat to coral reefs, are there any opportunities for synergies between your three recommendations and current measures?

Species traits: I cannot see where the species length and disc width values have been obtained for, please clarify it in this section. Also, I find confusing the sentence in lines 431-433, please clarify it.

Figure 4: The resolution of the sections c, d, g, and h hampers me from easily understanding what is being represented in that part. Since a similar image with better resolution is presented in Extended Data Figure 4, please consider redoing it.

I hope you find these suggestions useful and may help to improve the paper's clarity and quality.

Sincerely,

Imanol Miqueleiz

Reviewer #2 (Remarks to the Author):

This paper uses red list data to examine the extinction risk of coastal sharks and rays, compares these risks to other taxa, and identifies the likely drivers. The threats faced by sharks (and to a lesser degree rays) is relatively well established at this point and follows, for example, a global assessment of shark abundances. However, this paper tries to move beyond reducing abundances to think about extinction rates and true loss of biodiversity. To me, this is the weakness of the paper – despite decades of overfishing there are very few examples of actual marine species extinctions and this summary shows there are none for sharks and rays (Fig 1). Therefore, it is hard to believe that ~60% of species are truly threatened with extinction, especially with the increasing number of shark sanctuaries and protection of large remote areas. That is certainly not to say that sharks and rays aren't endangered and that local extirpation and functional extinction aren't likely or have already occurred, but that has been assessed by the MacNeil et al paper (and others) using actual survey data. Rather, this paper explicitly focuses on extinction rates and it is a huge step to go from Critically Endangered / Endangered / Vulnerable to extinct. Indeed the authors highlight this issue (p9-10) by pointing out that by assigning these categories across a species' range while it may be locally abundant (eg the abundance of sharks in French Polynesia means that any species occurring there are unlikely to ever go extinct). An assessment of sharks and rays in 'threatened' categories is still valuable, but may not be of sufficient interest to the journal (as opposed to discussing a potential extinction crisis). I also had a problem squaring risk with the species being in fewer countries – surely a large range means that the species is more likely to survive somewhere (only one country needs to be a shark sanctuary).

I am not very familiar with the red listing process, but suspect that it is done using a combination of data and expert opinion. It occurs to me that maybe - explicitly or implicitly – this expert opinion draws on species traits to assign a species to a category. If so, this may lead to some of the correlations discussed in the paper. Even if not, it would be worth discussing why this isn't the case to inform readers.

I wasn't convinced by the logic of why resident species might be less vulnerable than pelagic species (p7). Are resident species like whitetips really hiding within the reef? And even if they are that doesn't stop them being taken on hook and line – many groupers and snappers are reef-associated but are heavily overfished. Couldn't it just be an exposure of transients to coastal and pelagic fisheries and the lack of pelagic protected areas?

More minor points:

- Some of the text in the Introduction could use some editing. Is overfishing really a "silent killer"? There is a vast array of literature on this topic so is it really more "silent" (whatever that means) than climate change? And the loss of urchin predators leads to outbreaks of urchins in the Indian Ocean, not the loss of herbivores. There are also many more impacts than just on herbivores. And why give 4 examples of nutrient vectoring but none on their top down predation effects?
- Last sentence on p14 is awkwardly worded
- I wouldn't expect you to use irrelevant software (p15)... be good to give credit to the packages you used.

Reviewer #3 (Remarks to the Author):

- This manuscript very clearly contributes novel, important information regarding the conservation status of sharks and rays that inhabit coral reefs, particularly as it compares to the sweeping generalizations often touted about this group of species as whole. This differentiation is important, eye-opening, and will help managers worldwide as they move forward with conservation efforts across multiple scales. It is worthy of publication.

-Overall, methodology is sound, robust, and thorough in its description and execution. Figures are thoughtful in their presentation, easy to follow, and informative. Previous studies contributing data and methods have been appropriately cited. However, the manuscript could benefit from a more

detailed description of how the retrospective estimates of extinction risk were calculated for the Red List Index. It would be difficult to reproduce this work based on the current description alone.

- There are several minor grammatical errors that should be addressed - a few instances of missing article use with nouns, for example.

- In the "Species Traits Explain Extinction Risk" section, the authors state that Transient species had the highest overall threat level. While this is true, the authors might consider adding a brief explanation for the high proportion of Critically Endangered species in the Partial Residents category, as this runs counter to the overall generalization.

-The inclusion of National Attributes to explain extinction risk is an interesting choice and, I think, strengthens the authors argument for the need for implementation of regional level management strategies.

-Overall, I recommend publication with revisions.

Reviewer #1:

Dear authors,

I have read with high interest the manuscript “Half a century of rising extinction risk of coral reef sharks and rays”. The article provides a deep insight into the conservation status of coral reefs’ sharks and rays, and nice explanations about the drivers of the high extinction risk of this group. I will provide some comments on some parts of the text that could be enriched, and others that may need to be clarified, but in general I find the manuscript well written, interesting and with quality enough to recommend its publication in the journal with a minor revision of some aspects.

I first realized that the authors based this study, or at least part of it, in the previously published one, “Overfishing drives over one-third of all sharks and rays toward a global extinction crisis”. I would like to thank the authors for overcoming my initial skepticism, as I thought the paper would just be a repetition of the previous one with the focus on coral reef species. On the contrary, the manuscript deepens in the conservation status of these species and provides a broad and complete overview on the causes of the observed threat patterns. I would like to highlight how different sources of information, very different among them, have been used to create a text easily readable as a whole, well-structured and with a clear narrative that allows the reader to understand most of the results observed, and they were obtained.

Our response: Thank you for the kind words about our manuscript and the suggestions below that have helped us improve the clarity and understanding of our work.

I would like to point out some aspects that I consider would help to clarify the comprehension of the manuscript.

Line 88: I cannot find any rule against references in the abstract in the Author Guidelines, but they do not usually appear there, I leave it to the editor’s decision.

Our response and changes made to the MS: This article was originally submitted to a different journal where citations in the abstract were required but was subsequently transferred to *Nature Communications*. We have now removed these references from the abstract.

Lines 116-120: I struggle to see the connection between these two sentences. Could you explain better how eliminating large predators leads to large herbivores disappearing too?

Our response and changes made to the MS: We have reworded to explain that eliminating large predators leads to mesopredator release and this greater abundance is likely to affect the primary consumers resulting in herbivore disappearances.

Line 177 (and also lines 405-425): Where does this number come from? I assume you are using IUCN Red List habitat 9.8, but this search lead to 4,880 species. How did you increase this number up to 4,918 species?

Our response: We did search the IUCN Red List website for all species classified as occurring in habitat 9.8 (April 17, 2021). Since this search, there has been at least one update to the IUCN

Red List, which may change the results of the search as some species are removed due to taxonomic changes or some have updated habitat information. We further explain the addition of all Epinephelidae species despite their Red List Assessments (RLA) not all including 'coral reef' as habitat. We also added 20 species of sharks and rays that did not include 'coral reef' as a habitat on their RLA but we now know that they are likely to use these habitats (this is described in the "Species List and Attributes" section).

Changes made to the MS: Consequently, we have added a sentence in the methods section entitled, "Extinction Risk and Other Reef Taxa" and this sentence reads: "We included a total of 4,918 species (4,747 identified through the IUCN Red List website, with an additional 167 species due to including all Epinephelidae, and the 134 sharks and rays from this study)." The numbers are further broken down in Figure 1 where there is a column with the number of species for each group next to their respective bar.

Line 179: Eight species of delphinids, and also sirenids?

Our response and changes made to the MS: Yes, we have reverted to the common names "dolphin and sea-cow" to avoid confusion with the salamander family. Please note, sirenids are salamanders, not sea-cows, which are sirenians.

Lines 211-213: Filter feeders, that are the most threatened trophic group in figure 2c, are not "species of higher trophic position". It would be worthwhile to acknowledge this fact, considering the few filter feeders species, and their special condition as big and charismatic fishes (manta rays and whale shark), it is something that can be included.

Our response and changes made to the MS: This is correct that this group is also (highly) threatened, however, the results described are from an ordinal model where filter feeders were assigned the lowest number. In the MS, we have rephrased the sentence to state the model results and then explain why filter feeders do not follow this pattern: "The filter feeders (low trophic position) were threatened, likely due to other attributes including their transient nature and large size. Overall, however, the higher trophic level species were significantly more threatened, based on an ordinal model (z-value = 2.75, p = 0.006)."

Lines 227-228: This sentence is repeated from lines 219-220, it adds nothing new, please remove.

Our response and changes made to the MS: Thank you, removed.

Line 530: Maybe could be useful to include a table with species retrospective assessments as a Supplementary Material.

Our response and changes made to the MS: Good suggestion, have added this to Supplementary Table 2 with other species traits.

Extended data Figure 5: How were the quadrants delimited? Arbitrarily, mean or median values, other reason? Please explain it.

Our response: This is listed in the figure caption - “Dotted lines indicate the median values of Sightings Per Unit Effort (SPUE) and Threat Percentage.”

Changes made to the MS: We have also added labels next to the lines on the figure delineating the quadrats to clarify.

Discussion: I enjoy the discussion of protective measures to prevent overfishing in coral reefs. How could these measures potentially be combined with the ones that are currently been applied? Since climate change has been traditionally classified as the main threat to coral reefs, are there any opportunities for synergies between your three recommendations and current measures?

Our response: Thank you, our three recommendations are currently being used around the world, to varying degrees, however, we reiterate that these need to be implemented quickly and at a broad-scale to halt declines and initiate recovery. The first two recommendations would also serve as appropriate actions to adapt to the effects of climate change, both by improving coral reef resilience through biodiversity preservation, and at a socioeconomic level through alternative livelihoods leading to alternative income streams for those that rely on reef resources.

Changes made to the MS: We have added the following text after the recommendations in the discussion: “These recommendations are currently being applied to varying degrees and with varying success. The first two also address climate change threats to coral reefs by increasing resilience by maintaining abundance and conserving biodiversity, and reducing human reliance on reefs through alternative livelihoods.”

Species traits: I cannot see where the species length and disc width values have been obtained for, please clarify it in this section. Also, I find confusing the sentence in lines 431-433, please clarify it.

Our response and changes made to the MS: We have added a sentence in the “Species List and Attributes” section stating “Maximum size values were obtained from the published species’ IUCN Red List Assessments.”

Have reworded lines 431-433 to better explain: “As shark sizes are measured as total length and rays are generally measured as the disc width, models were tested both with and without maximum sizes nested in measurement type (i.e., total length and disc width).”

Figure 4: The resolution of the sections c, d, g, and h hampers me from easily understanding what is being represented in that part. Since a similar image with better resolution is presented in Extended Data Figure 4, please consider redoing it.

Our response: This diminution of resolution most likely occurred through the submission process and is out of the control of the authors. Please either email us or the editor for the original higher resolution versions of the figures.

I hope you find these suggestions useful and may help to improve the paper's clarity and quality.

Our Response: Thank you for your comments and suggestions that undoubtedly have improved the manuscript clarity.

Sincerely,
Imanol Miqueleiz

Reviewer #2 (Remarks to the Author):

This paper uses red list data to examine the extinction risk of coastal sharks and rays, compares these risks to other taxa, and identifies the likely drivers. The threats faced by sharks (and to a lesser degree rays) is relatively well established at this point and follows, for example, a global assessment of shark abundances. However, this paper tries to move beyond reducing abundances to think about extinction rates and true loss of biodiversity. To me, this is the weakness of the paper – despite decades of overfishing there are very few examples of actual marine species extinctions and this summary shows there are none for sharks and rays (Fig 1).

Our Response: While there are no known extinctions for coral reef sharks and rays, three elasmobranch species are assessed as Critically Endangered (Possibly Extinct) resulting in an extinction rate similar in magnitude to terrestrial vertebrate biodiversity (Dulvy et al. 2021). We appreciate that the referee may not have had a chance to see this recent result or keep track of the burgeoning literature showing numerous widespread local and regional extinctions and the scale and extent seems to grow as evidence is accumulated (e.g. see the papers on sawfishes and angel sharks by Yan et al. 2021 and Lawson et al. 2020, respectively). Aside from our work, there is an excellent paper (Webb and Mindell 2015) that shows that the apparent paucity of marine extinctions is due to an understudy bias, that should be resolved as IUCN Red Listing ramps up. Our paper shows the effect of more intensive Red Listing, confirming the Webb & Mindell finding that the marine extinction rate is likely similar to the terrestrial rate (with 20-25% of species threatened). Indeed, our study shows that for some groups, such as coral reef sharks and rays, the prognosis is much much worse.

Similarly, there are localized functional extinctions noted in several studies including the MacNeil study. Although this may not be a full extinction, we show that severe declines have occurred in just the past half century, during which the human population has doubled. The dramatic decline in abundances observed in these years does lend credence to the high likelihood of marine species extinctions in the near future if these patterns are allowed to continue. Considering elasmobranchs represent the earliest extant jawed vertebrate lineage, the extreme depletion in just 50 years is alarming and is consistent with a greater than background rate of extinction risk.

Changes made to the MS: To stress the quick timeline of decline, we rephrased the sentence in the discussion where overfishing is mentioned as the major threat to add the second clause: “Overfishing is the major threat reported and the main cause for population declines, *causing dramatic declines in a very short space of time (i.e., the past 50 years).*”

Therefore, it is hard to believe that ~60% of species are truly threatened with extinction, especially with the increasing number of shark sanctuaries and protection of large remote areas.

Our Response: The IUCN Red List addresses the probability of extinction. It is the most broadly used estimator of extinction risk and has set quantitative criteria which are uniformly applied across all taxa. We assessed the relative extinction risk from Least Concern to Critically Endangered (possibly Extinct) of all 1,199 shark and ray species before subsetting out this dataset for analysis. Assessments are reviewed by a minimum of two experts to ensure the criteria have been properly applied to the species being assessed. Threatened with extinction means (under the IUCN A criterion) that a species is declining at a rate of 30% over the span of three generation lengths - if this decline goes unmanaged and unhalted, it is incontrovertible that the species will go extinct. At present there is no single threatened reef elasmobranch species that is adequately managed throughout its geographic range and is thus safe from extinction.

Unfortunately this is not widely understood by the scientific community and reported in the literature, however, shark ‘sanctuaries’ are a misnomer created by one NGO. These ‘sanctuaries’ do not meet any IUCN protected area definition and, with few exceptions, they are not resourced or enforced. Consequently, despite the large number of ‘sanctuaries’, there continues to be declines in abundances of shark and ray populations. Most ‘sanctuaries’ are ‘paper parks’ and not well-enforced (Pieraccini et al. 2017; Pereira da Silva, 2019), not large enough to have a noticeable effect for sharks (Dwyer et al. 2020), not part of a well-connected MPA network (Martín et al. 2020), continue to be open to fishing where sharks must be returned (despite high post-release mortality of some species)(Albano et al. 2021), or some combination of the above. Protected areas in remote locations do in fact serve as some of the last refuges for these species (Letessier et al. 2019). However, many reef species beyond the photogenic few do not make large-scale migrations, therefore, these parks will not provide ‘spillover’ to coastal areas which are most at-risk to overfishing. We do discuss that MPAs with proper implementation, including the factors mentioned above, should be the primary tool used to improve coral reef shark and ray status (Lines 345-352).

That is certainly not to say that sharks and rays aren’t endangered and that local extirpation and functional extinction aren’t likely or have already occurred, but that has been assessed by the MacNeil et al paper (and others) using actual survey data.

Our Response: The MacNeil paper does not assess extinction risk of reef species, rather abundances at local reefs and compares this value to the mean for each species across all sampling sites. As such, the MacNeil paper is about abundance relative to the global average, but it is hard to translate this value, on its own, into something more meaningful for conservation assessment. The IUCN Red List Assessments (RLAs) quantify species’ global extinction risk based on the rate and extent of decline scaled according to life history, rather than a localized

area. RLAs use real data, including the results of the MacNeil study (i.e. *Triaenodon obesus* and *Taeniura lymma*), to determine risk across the entire range of a species, rather than just the small portion of their range that can be reasonably sampled. Bear in mind the MacNeil paper only considered a narrow depth slice and very specific habitat of these species' global ranges.

Rather, this paper explicitly focuses on extinction rates and it is a huge step to go from Critically Endangered / Endangered / Vulnerable to extinct. Indeed the authors highlight this issue (p9-10) by pointing out that by assigning these categories across a species' range while it may be locally abundant (eg the abundance of sharks in French Polynesia means that any species occurring there are unlikely to ever go extinct). An assessment of sharks and rays in 'threatened' categories is still valuable, but may not be of sufficient interest to the journal (as opposed to discussing a potential extinction crisis). I also had a problem squaring risk with the species being in fewer countries – surely a large range means that the species is more likely to survive somewhere (only one country needs to be a shark sanctuary).

Our response: Global extinction is a product of multiple local extinctions summing to regional extinctions until all regions are affected and the global extinction occurs (and is eventually detected). All these stepping stones toward extinction can be illustrated by the various species in the panels of figure 2 of Dulvy et al. 2021 *Curr. Biol.* 31, 4773–4787. Conceptually, the leap from Critically Endangered to Extinct seems a long way. Digging deeper into the criteria, however, please note the IUCN A criterion compresses the extinction scale based on declines from 0% to 100%, with IUCN category thresholds at 30, 50 and 80% declines (over the greater of 10 years or 3 generation lengths). Further, this decline rate criterion is augmented by the new threat model to infer Possible Extinct and Extinct species (see IUCN Red List Guidelines 2019, P83-87 and three associated papers). So conceptually the step from rarity to extinction might seem large to the referee, however, the criteria narrows the scope of the problem.

While it is true that wide-ranging species do have a higher likelihood of occurring in abundance at some locations, they are also exposed to a broader range of management jurisdictions. Additionally, depending on the seasonal migratory or diel movement pattern of the species, when individuals leave the location or country with protection, they will be at-risk of capture and reduce the population of the 'shark sanctuary' country. So a handful of moderately effective shark 'sanctuaries' may not be enough to avoid continuing global decline, and certainly not for rays, which are not subject to regulation in shark 'sanctuaries'.

I am not very familiar with the red listing process, but suspect that it is done using a combination of data and expert opinion. It occurs to me that maybe - explicitly or implicitly – this expert opinion draws on species traits to assign a species to a category. If so, this may lead to some of the correlations discussed in the paper. Even if not, it would be worth discussing why this isn't the case to inform readers.

Our Response: The short answer is that a lot more than life history traits goes into an assessment. The longer answer is the IUCN Red List has set quantitative criteria which are applied in order to assign a category to a species. Therefore, species are not assessed based on their traits, rather an understanding of their population trend due to habitat loss, fishing, and other threatening processes. In the event that there is no observed, estimated, projected,

inferred, or suspected information about a species' population trend, it would be assessed as Data Deficient, regardless of size, location, or other intrinsic traits.

Red List Assessments (RLAs) are completed using available data, which may include proxy data (fisheries landings or absence thereof) and expert understanding of these data (historical ecology). Traits are considered through generation length, which is used to scale annual population decline rates to the productivity of a species (r_{max} is highly correlated with generation length). And for this reason we tend not to bring generation length into these correlational searches. Body size is not directly considered in RLAs, however, size, habitat use, depth range, geographic range, diel and migratory behaviour (among other things) may enter due to their influence on catchability resulting in a higher likelihood of being caught in a larger suite of fishing gears, as well as larger-bodied species tend to be intrinsically more at risk. However, body size is but one of at least three dimensions of life histories, including time-related traits, and reproductive allocation.

I wasn't convinced by the logic of why resident species might be less vulnerable than pelagic species (p7). Are resident species like whitetips really hiding within the reef? And even if they are that doesn't stop them being taken on hook and line – many groupers and snappers are reef-associated but are heavily overfished. Couldn't it just be an exposure of transients to coastal and pelagic fisheries and the lack of pelagic protected areas?

Our Response: We appreciate that Whitetip Reef Shark is one of the most widely and frequently observed reef shark (in the Indo-Pacific), however this species is actually the exception. Most resident species are rays or small benthic sharks that hide within reef structures (e.g Coral Cat Sharks, Bamboo Sharks, and Maskrays). Further, these species are unlikely to be caught in high abundances on hook and line because they feed mainly on invertebrates.

Additionally, please note the language used in the paper. We do not compare resident and pelagic species. Instead, we compare resident, partial resident, and transient species. Transient species are not all pelagic and include those with a wide range of habitats, therefore, the lack of pelagic protected areas is of limited concern to this group. We do make the point that transient species are exposed to fisheries outside of reef habitats, which we hypothesize has led to their higher threat status (lines 213-216).

Changes made to the MS: We have added a clause to the sentence in line 213 to explain resident species are primarily benthic: "Resident species, *including mainly benthic sharks and rays*, often hide within the structures on a coral reef and are not easily caught by fishing gears; as opposed to partial residents or transient species, which transit through soft-bottom habitats which are comparatively easy locations to operate trawls and gillnets effectively."

More minor points:

- Some of the text in the Introduction could use some editing. Is overfishing really a "silent killer"? There is a vast array of literature on this topic so is it really more "silent" (whatever that

means) than climate change? And the loss of urchin predators leads to outbreaks of urchins in the Indian Ocean, not the loss of herbivores. There are also many more impacts than just on herbivores. And why give 4 examples of nutrient vectoring but none on their top down predation effects?

Our Response and changes made to the MS: We have retained its use because one cannot directly observe the effect of fishing on shark populations, and there are few fishing gradients along which the effects can be observed. With regard to the urchin outbreaks, we restructured the sentence to clarify that mesopredator release disrupts ecosystem function: “Such functional losses distort ecological pyramids through mesopredator release⁹, leading to ecosystem disruptions⁸ and cascading disappearances of functionally important herbivores¹⁰, which can lead to outbreaks of reef-degrading echinoderms.”

Our fourth example of nutrient vectoring does address top-down predation effects. The example being Tiger Shark mediating foraging of seagrass beds. Previous studies have shown that the behaviour effect from predator presence can have a stronger effect on prey than direct predation. However, we had added this at the end to clarify: “Finally, larger transient apex predators, like the tiger shark (*Galeocerdo cuvier*) mediate ecosystem structure and function of seagrass beds through fear-induced changes in grazing behaviour of turtles and dugongs^{19,23}, which may be a stronger effect than direct predation²⁴.”

- Last sentence on p14 is awkwardly worded

Our Response: We have reworded this sentence.

- I wouldn't expect you to use irrelevant software (p15)... be good to give credit to the packages you used.

Our Response: We name and reference the packages throughout the rest of the statistics section. It seemed best to have the package name and citation after the sentence describing the test completed rather than listed at the top and restated.

Reviewer 3

- This manuscript very clearly contributes novel, important information regarding the conservation status of sharks and rays that inhabit coral reefs, particularly as it compares to the sweeping generalizations often touted about this group of species as whole. This differentiation is important, eye-opening, and will help managers worldwide as they move forward with conservation efforts across multiple scales. It is worthy of publication.

Our response: Thank you!

-Overall, methodology is sound, robust, and thorough in its description and execution. Figures are thoughtful in their presentation, easy to follow, and informative. Previous studies contributing data and methods have been appropriately cited. However, the manuscript could benefit from a more detailed description of how the retrospective estimates of extinction risk

were calculated for the Red List Index. It would be difficult to reproduce this work based on the current description alone.

Our response and changes made to the MS: Thank you. We have added more details in how the retrospective Red List categories were assigned: “In particular, we used reconstructed catch data from the Sea Around Us, which includes the dates, locations, and species groups exploited. With this information, along with an understanding of species-specific traits and response to fishing, we could estimate that if fishing pressure increased throughout the 1970s and reached a peak in the early 2000s, the species in question was likely Least Concern in 1970, possibly Near Threatened by 1980, and likely in a threatened category by 2005. We built a picture of the history of fishing pressure using the Sea Around Us data and worked backwards from the current category to determine if the previous category (e.g., from 2020 to 2005, then 2005 to 1980, then 1980 to 1970) was likely to be: (i) the same, (ii) better by one, or (iii) two categories. Thus, the range of possible choices is highly constrained at each timestep.” Additionally, we have added the retrospective Red List categories for each species to Supplementary Table 2 at the suggestion of another reviewer.

- There are several minor grammatical errors that should be addressed - a few instances of missing article use with nouns, for example.

Our response: We have gone through the manuscript thoroughly to check for grammatical errors.

- In the "Species Traits Explain Extinction Risk" section, the authors state that Transient species had the highest overall threat level. While this is true, the authors might consider adding a brief explanation for the high proportion of Critically Endangered species in the Partial Residents category, as this runs counter to the overall generalization.

Our response and changes made to the MS: This is a good point. We have added to the section to explain this pattern: “Partial residents did have a higher proportion of Critically Endangered species than the other two groups. This may be due to other traits, as the CR partial resident species tended to be larger bodied species with long generation lengths or occurred in less than five nations.”

-The inclusion of National Attributes to explain extinction risk is an interesting choice and, I think, strengthens the authors argument for the need for implementation of regional level management strategies.

Our response: Thank you!

-Overall, I recommend publication with revisions.

REVIEWER COMMENTS

Reviewer #1 (Remarks to the Author):

Dear authors,

Thank you for the explanations provided in the comments. Since all my questions and doubts have been solved, I no longer have any further requirements for the authors. Thus, I am glad to recommend the manuscript for its acceptance in the journal.

Reviewer #2 (Remarks to the Author):

Thank you to the authors for their responses. I want to re-iterate that I do not doubt that many shark populations are declining rapidly and conservation measures are urgently needed. Equally, the red listing process has been an enormous undertaking and it is important to highlight the trends within the larger data set, as the authors do here for coral reef species. Rather, my comments related to the headline finding and message that “Nearly two-thirds (59%) of the 134 coral-reef associated shark and ray species are threatened with extinction.”. The paper clearly lays out how this number is derived and it fits with the IUCN red listing framework. But despite this, I have a hard time seeing how a species like *Carcharhinus perezi* is really at risk of complete extinction when it is relatively site attached (Dwyer et al 2020) and populations appear stable and reasonably high (eg Clementi et al 2021, Talwar et al 2020) in a large archipelago like the Bahamas where shark fishing is banned. But I can also see why the species is classified as endangered across its entire range and thus it fulfils the criteria of being counted as threatened with extinction. I also still feel that this problem particularly applies to the 30% of ‘Vulnerable’ species that represent 50% of the headline message, but again seem a long way from losing every single individual across their entire range.

But I note the other reviewers don’t share this issue and the paper is consistent with the red list framework that lists these categories as ‘threatened with extinction’. Thus the paper should move forward with the findings, and readers can see all the details in the text. Personally I would like to see these nuances of heterogeneity within the species’ range discussed in the Discussion (currently only 2 paragraphs), but at this point I will leave that decision to the editor.

One minor comment. I still have an issue with “Such functional losses distort ecological pyramids through mesopredator release⁹, leading to ecosystem disruptions⁸ and cascading disappearances of functionally important herbivores¹⁰, which can lead to outbreaks of reef-degrading echinoderms^{11,12}.”. The evidence of the top-down effects of high trophic levels on ecosystem disruptions is more mixed than indicated, the Rasher paper does not suggest the disappearance of herbivores (rather they reallocate their space use), and this sentence still suggests that the loss of herbivores leads to outbreaks of echinoderms, which is not what those citations suggest.

Reviewer #3 (Remarks to the Author):

Revisions made to the manuscript have adequately addressed my concerns as a reviewer.

Reviewer #1 (Remarks to the Author):

Dear authors,

Thank you for the explanations provided in the comments. Since all my questions and doubts have been solved, I no longer have any further requirements for the authors. Thus, I am glad to recommend the manuscript for its acceptance in the journal.

Response: Thank you for your helpful comments.

Reviewer #2 (Remarks to the Author):

Thank you to the authors for their responses. I want to re-iterate that I do not doubt that many shark populations are declining rapidly and conservation measures are urgently needed. Equally, the red listing process has been an enormous undertaking and it is important to highlight the trends within the larger data set, as the authors do here for coral reef species. Rather, my comments related to the headline finding and message that “Nearly two-thirds (59%) of the 134 coral-reef associated shark and ray species are threatened with extinction.”. The paper clearly lays out how this number is derived and it fits with the IUCN red listing framework. But despite this, I have a hard time seeing how a species like *Carcharhinus perezii* is really at risk of complete extinction when it is relatively site attached (Dwyer et al 2020) and populations appear stable and reasonably high (eg Clementi et al 2021, Talwar et al 2020) in a large archipelago like the Bahamas where shark fishing is banned. But I can also see why the species is classified as endangered across its entire range and thus it fulfils the criteria of being counted as threatened with extinction. I also still feel that this problem particularly applies to the 30% of ‘Vulnerable’ species that represent 50% of the headline message, but again seem a long way from losing every single individual across their entire range.

But I note the other reviewers don’t share this issue and the paper is consistent with the red list framework that lists these categories as ‘threatened with extinction’. Thus the paper should move forward with the findings, and readers can see all the details in the text. Personally I would like to see these nuances of heterogeneity within the species’ range discussed in the Discussion (currently only 2 paragraphs), but at this point I will leave that decision to the editor.

Response: Thank you. The Editor has suggested that based on these comments that we consider two issues – (1) moderating some of our conclusions, (2) expanding the Discussion to detail the heterogeneity in status. In relation to (1), we have changed the wording in a few places within the text to achieve this moderation (e.g. line 93 added “Here we show that...” and line 264 changed “national extinction risk” to “national percentage of globally threatened species”). In relation to (2), a significant amount of information and analysis on the heterogeneity of status is provided in the section entitled “Where Are Reef Sharks Threatened?” We believe that the information presented there, including some discussion of this topic and its consequences is sufficient for the reader to understand this issue.

One minor comment. I still have an issue with “Such functional losses distort ecological pyramids through mesopredator release⁹, leading to ecosystem disruptions⁸ and cascading disappearances of functionally important herbivores¹⁰, which can lead to outbreaks of reef-degrading echinoderms^{11,12}.”. The evidence of the top-down effects of high trophic levels on ecosystem

disruptions is more mixed than indicated, the Rasher paper does not suggest the disappearance of herbivores (rather they reallocate their space use), and this sentence still suggests that the loss of herbivores leads to outbreaks of echinoderms, which is not what those citations suggest.

Response: The reviewer is correct – some aspects of this sentence do not exactly match the information from the papers cited. As such we have revised the sentence to remove the reference to the outbreaks of echinoderms, and have used a different reference (Ruppert et al. 2013) about changes to coral reef food webs and the abundance of herbivores. The new sentence now reads: “ Such functional losses distort ecological pyramids through mesopredator release (Sherman et al 2019), leading to ecosystem disruptions (Graham et al 2017) and cascading changes down coral reef food chains that lead to declines in functionally important herbivores (Ruppert et al. 2013).” [note that we have shown the citation with primary author and year rather than a numbered reference for clarity here].

Reviewer #3 (Remarks to the Author):

Revisions made to the manuscript have adequately addressed my concerns as a reviewer.

Response: Thank you for your helpful comments.